



# Cambial-age related correlations of stable isotopes and tree-ring widths in wood samples of tree-line conifers

Tito Arosio[1,2], Malin M. Ziehmer-Wenz[1,2,5], Kurt Nicolussi[3], Christian Schlüchter[4,2], Markus Leuenberger[1,2]

[1]Climate and Environmental Physics, Physics Institute, University of Bern, 3012 Bern, Switzerland
[2]Oeschger Centre for Climate Change Research, University of Bern, 3012 Bern, Switzerland
[3]Institute of Geography, University of Innsbruck, 6020 Innsbruck, Austria
[4]Institute of Geological Sciences, University of Bern, 3012 Bern, Switzerland
[5]Swiss Tropical and Public Health Institute, 4051 Basel, Switzerland

*Correspondence to*: Tito Arosio (tito.arosio@climate.unibe.ch)

**Abstract.** A recent analysis of stable isotopes of the Alpine Holocene Tree-Ring Dataset, consisting of samples from 192 larch and cembran pine trees, revealed that $\delta D$ and $\delta^{18}O$ exhibit no trends in adult trees, but evidence trends in the juvenile period of the first 100 years of cambial age. In this work we applied the Spearman statistical analysis on different cambial age classes

to verify if these changes were correlated with tree-ring width values, that are known to show age trends. The results prove a significant correlation between tree-ring-width (TRW) and both hydrogen and oxygen stable isotopes before 100 year of cambial age, but not afterwards, in both larch and cembran pine. A trend in the correlation values was also found between the two water isotopes, while no trend was found in correlations involving $\delta^{13}C$. We hypothesized the $\delta D$ and $\delta^{18}O$ values reflect the higher xylogenesis activity of the juvenile period, that is associated with reduced atom exchanges of photosynthates with

xylem water. The result indicates that the climate response of $\delta D$ and $\delta^{18}O$ may differ in the juvenile and mature period of tree life at treeline.

## 1 Introduction

Width values of tree rings (TRW) are widely used for paleoclimatic reconstruction with the major advantage of having annual resolution (Fritts 1976, Büntgen et al. 2020a, Loader et al. 2020). The fractionation of the stable isotopes of hydrogen, oxygen

and carbon in the tree-ring cellulose have also been largely used as paleoclimate proxies (McCarroll and Loader 2004). An issue of stable isotopes for paleoclimatic studies is the possible presence of cambial age effects, that has been studied in various recent papers with conflicting results (Büntgen et al. 2020b; Esper et al. 2010; Helama et al. 2015; Xu et al. 2020). We have recently shown that in our Alpine Holocene Tree Ring Dataset, the age-trends of $\delta D$, $\delta^{18}O$ and $\delta^{13}C$ are present in the juvenile phase in the first 100 cambial years, but they are absent afterwards, with differences in the two species in the juvenile phase.

(Arosio et al. 2020a). The dataset covering the last 9,000 years is large, based on almost 200 trees of two conifer species (Larix decidua and Pinus cembra) and cambial ages ranging from 1 to 700 years. It offers an opportunity not only to analyse trends

in the stable isotopes themselves (Arosio et al. 2020a) but also to investigate the relationship between TRW and isotopes along the cambial age. This would allow us to study the causes for the deviating stable-isotope trends of the juvenile phase, their possible physiological mechanisms and their effects on climate responses that are still unclear. Only a few studies analysed the correlations among TRW values and stable isotope fractionation with rather contrasting results. Most of them used detrended TRW values and the calculated values of the correlation factors were generally in the range 0.2 - 0.3 (Hafner et al. 2011; Kirdyanov et al. 2008; Schollaen et al. 2013; Shestakova et al. 2019; Weigl et al. 2007), but one work found the correlations not significant (Sidorova et al. 2010). However, none of the studies has taken into consideration how the cambial age of isotope samples can affect such correlation analyses. To fill this gap, we reanalyzed our database and divided the samples in classes of cambial age, with attention to the juvenile phase, i.e. the first 100 cambial years, and we studied the correlations between TRW and stable isotopes, using raw data as well as detrended data applying different detrending methods.

## 2 Material and method

The tree-ring database used in this work as well as the analysis and the estimation of the cambial age were described before (Arosio et al. 2020a). The samples come from the Eastern Alpine Conifer Chronology (EACC) (Nicolussi et al. 2009) using two species: the deciduous larch (Larix decidua Mill.) and the evergreen cembran pine (Pinus cembra L.). Wood samples spanning 5 years of tree-rings have been prepared and analyzed for stable isotope ratios as described in recent publications (Arosio et al. 2020a; Ziehmer et al. 2018). The procedure of cellulose extraction (Boettger et al. 2007) and the triple-isotope analysis are described in (Filot et al. 2006; Loader et al. 2015; Ziehmer et al. 2018). The results are reported in per mil (‰) relative to the Vienna Pee Dee Belemnite (VPDB) for carbon and to Vienna Standard Mean Ocean Water (VSMOW) for hydrogen and oxygen (Coplen 1994). The precision of the measurement is ±3.0‰ for hydrogen, ±0.3‰ for oxygen and ±0.15‰ for carbon (Loader et al. 2015).

### 2.1 Correlation analysis:

All the tree-ring and cellulose isotope data were first divided in 5 classes of cambial age (1-25, 26-50, 51-75, 76-100, 101-200, 201-300, >300 year) for the two species, Table 1 displays the number of the samples and the number of the trees for each class. The correlations between the raw TRW and the isotopes values were studied. To avoid a geographical effect that can lead to artificial trends, we have normalized the isotope values of each tree by subtracting the tree mean from each value.
The correlations between TRW and the isotope values were calculated using the Spearman's rank correlation coefficient for each age class and also for the total data of each species (indicated by the blue boxes in the graphs). We chose the Spearman method (Zar 1972) because it is widely used when comparing data with non-normal distribution (e.g. TRW data) with normal distributed data (e.g. the isotope values). The p-values (statistical significance) were calculated for each correlation and are displayed as bars in Figure 1 and 2. If $P < 0.05$, they are labeled as "ns" (not significant) in the graphs. To identify possible correlations derived by common trends, we calculated the correlations in three different scenarios: (i) by using the raw values



of isotopes and TRW (Fig. 1 a,b,c and 2 a,b,c); (ii) by applying a linear detrend to the isotope and TRW values, and removing linear trends in each age class. (Fig. 1d,e,f and 2d,e,f); and (iii) by using the isotope values and applying a spline detrend to all

TRW values not subdivided into classes(Fig. 1g,h,i).

Spline detrending is a common methodology used in dendroclimatology to remove the ageing signal. In this study we calculated it by applying a linear spline detrending function, using the defaults sets, to the raw TRW utilizing the R library dplR (Bunn 2008).

For further confirmation on the stability and robustness of Spearman correlation coefficient, we run a Monte Carlo simulation

with random subsampling of the samples of half of the total population and calculating the Spearman's correlation factor in a loop of 5000 cycles. The results are then compared with the calculated Spearman correlation results for the total population. To assess the stability of the correlations of the number of randomly selected samples from the total population, we also run the Monte Carlo simulation for each class and species.

**Results**

The r values of correlation between TRW and the three isotopes in larch (red) and cembran pine (turquoise) at the various cambial ages are shown in Figure 1. The upper plots (Fig. 1a, b, c) show the correlations between the raw TRW data, while the middle plots (Fig. 1d, e, f) show the correlations between the TRW and the isotopes values that underwent linear detrending. The lower plots (Fig 1g, h, i) show the correlations in which the TRW values were detrended with a spline function. The blue boxes indicate the correlation r values of all trees, using all tree samples without cambial age class discrimination.

The correlations between TRW and $\delta$D (Fig. 1a) are highly significant and negative in both species up to 50 years of cambial age and then they tend to decrease and become non-significant after 100 years of age, and for larch the values become even positive. The linear detrending of TRW of each class (Fig. 1d) did neither change the r values nor the trends of the correlations in the cambial age classes.

Similar is the pathway for correlations between TRW and $\delta^{18}$O (Fig. 1 b,e) with positive and significant correlations in the

first 50 years followed by a loss of significant correlation after 100 cambial years. In both cases the correlations TRW-$\delta$D and TRW-$\delta^{18}$O of the two youngest cambial age classes presented absolute values that are much higher with respect to those using all values as shown in the blue box.

The correlations between TRW and $\delta^{13}$C (Fig. 1 c, f) do not present evident changes in the cambial age classes, the correlations are non-significant with the exception of the class of 50 years of cembran pine that has a correlation r value close to 0.3.

The lower plots that use spline-detrended TRW values (Fig. 1 g, h) display trends different from the ones above. The correlations between TRW and deuterium and those between TRW and oxygen show an increase of non-significant values; the second age class (26-50 year) of cembran pine is the one with the absolute highest correlation value in both isotopes. In larch the first two classes are non-significant, in both isotopes afterwards, r is in the order of the correlation when all the values are considered (blue box). The correlations between the detrended TRW and $\delta^{13}$C are generally significant including the r of

all values (Fig. 1 i), with a pattern that is different from those of the raw and linear detrended TRW above.





We applied the same approach to study the correlations between the isotopes themselves, firstly using the raw data (Fig. 2 a, b, c) and then using linear detrended data to each class (Fig. 2 d, e, f). The correlations between δD and δ¹⁸O have different behaviors in the two species. The youngest age class for cembran pine show negative and significant r values, after 75 year they are non-significant or close to 0. No difference is evident between the plots of the raw data and the linear detrended ones,

except for non-significant correlations of the total and 75 year class of cembran pine (Fig. 2a vs 2d). An opposite trend is evident for larch, where the younger classes have low and non-significant r values, followed by classes with increased and significant r values, no difference is evident between the raw data and the linear detrended classes. The correlations between δ¹³C and δ¹⁸O and between δ¹³C and δD do not show any evident trend with cambial age.

The results of Monte Carlo simulations for each age class of the two species are shown in the supplementary material. The

frequency of the correlation coefficients for randomly selected samples from half of the total population has always a normal distribution around the observed correlation value. This fact proves the robustness of correlations in relation to possible random sub-samples for this case. We observe that the decrease in sample size may spread the range of correlation coefficient frequency and lead to a non-normal distribution. Based on these calculations, we found a threshold of ~30 samples above which the correlation strength is close to that of the total population.

**Discussion:**

The aim of the work was to verify if the correlations among the tree-ring parameters (TRW and stable isotopes) are affected by the cambial age, in particular in the juvenile phase (Arosio et al. 2020a). The results of this study show that there is a moderate but significant correlation between TRW and the water isotopes (δD and δ¹⁸O) before 100 year of cambial age, and that after that the correlation values decline and become non-significant. The Monte Carlo simulations confirmed that

correlations although moderate were robust and stable and that a population of at least 30 samples is necessary to draw robust conclusions, otherwise correlations may not be representative. The correlations were positive with oxygen isotope, while they were negative with the hydrogen isotope. This occurred both in larch and in cembran pine with similar behaviour and values, indicating that the trends are not species-specific (Fig. 1). However, a major difference between the two species was found in the correlations between δD and δ¹⁸O with significant positive correlation factors for larch after 50 years, and significant

negative values for cembran pine only before 100 years (Fig. 2). The correlation values between TRW and δ¹³C were generally low and without evident trends related to cambial age. Moreover, we found that linear detrending of each age class did not have a major effect on the correlation values, while the spline detrending of TRW strongly reduced the values. An additional result was that the correlation between the TRW and the two water isotopes of all the trees, not divided in cambial age classes, were non-significant. This observation partially explains the contradictory published reports on the correlations (Hafner et al.

2011; Kirdyanov et al. 2008; Schollaen et al. 2013; Shestakova et al. 2019; Weigl et al. 2007; Sidorova et al. 2010), since their tree ages were not given. These new findings are in good agreement with our recent work showing that in the trees of the same dataset the cellulose δ¹⁸O and δD have age trends before but not after 100 cambial years, with the exception of a minor trend





of δD in larch (Arosio et al. 2020a), which are consistent with most of previous works on conifers (Daux et al. 2011; Kilroy et al. 2016; Klesse et al. 2018; Lipp et al. 1993).

The significant correlations between TRW and water isotopes in juvenile age may be due to biological processes that change during aging. An important process is wood formation (xylogenesis) that is faster and occurs over a longer period of the year in the juvenile phase, i.e. before 100 years, and is characterized by wider tree-rings (Li et al. 2013; Rossi et al. 2008). The TRW have a maximum width around an age of 30 cambial years, followed by a negative exponential decrease (Bräker 1981). It was shown that with a different cambial activity regime the fractionation of δD and $\delta^{18}O$ in cellulose changes due to isotopic

exchange with xylem water. During rapid growth glucose produced by photosynthesis is rapidly transformed into sucrose and used for cellulose synthesis (Szejner et al. 2020). This reduces the possibility to exchange the $^{18}O$ of glucose with non-enriched xylem water (Farquhar 1998; Sternberg et al. 2006). The same, but with opposite sign, occurs with the carbon-bound hydrogen atoms that are highly δD depleted in the photosynthetic glucose and that exchange with the δD xylem water, leading an enrichment of δD (Augusti et al. 2006). Thus, a faster growth is accompanied by wider rings, more enriched $\delta^{18}O$ and more

depleted δD, partially explaining the correlations we observed. This process is particularly relevant for the juvenile stage when the rings are wider.

This approach may also explain the lack of correlations between TRW and $\delta^{13}C$ (Fig. 1c), since carbon does not exchange isotopes with xylem water. It explains also why spline detrending of TRW reduced the correlations, since it removed growth-intensity signal. This lack of significant correlations between TRW and $\delta^{13}C$ during tree life may also indicate that the $^{13}C$

depleted carbon stored in starch (Tcherkez et al. 2011) does not become more important in the adult phase with respect to the juvenile phase, and that there is no direct link between its usage and the cambial activity.

The correlation, between δD and $\delta^{18}O$, changed during the juvenile phase in both phases with parallel trends is a novel finding and difficult to interpret at present, particularly because of the lack of understanding of biological fractionation of hydrogen in larch. What is common to the two species is the gradual transition from 1 to 100 cambial years of correlations among water

isotopes, which stabilizes after 100 years.

The particularity of δD in larch cellulose is described before (Arosio et al. 2020b), but without understanding what specific metabolic step is involved, and how this can change in relation to metabolism.

In conclusion, our results confirm the existence of a juvenile phase in the δD and $\delta^{18}O$ isotopes of cellulose that we attribute to a high cambial activity that decreases with higher age. This indicates a different climatic sensitivity of different age groups,

in accordance with (Xu et al. 2020), implying that a simple detrending of the values of water isotopes may not be sufficient for paleoclimatic studies.

**Acknowledgments.**

We are grateful to Peter Nyfeler for the precious assistance during stable isotopes measurements, to Andrea Thurner and Andreas Österreicher for the preparation of the isotope samples from Alpine sites and the the civil service collaborators: Lars

Herrmann, Giacomo Ruggia, Jonathan Lamprecht, Yannick Rohrer, Rafael Zuber. This work was supported by the Swiss





National Science Foundation projects (SNSF, 2000212_144255, 200020_172550) as well as by the Austrian Science Fund (FWF, grant I-1183-N19) and is supported by the Oeschger Center for Climate Change Research, University of Bern, Bern, Switzerland (OCCR).

**Competing interests.**

The authors declare that they have no conflict of interest.

**Author contributions.**

TA and MMZ performed the stable-isotope analyses. TA drafted the first version of the manuscript. KN collected the samples and performed the cross-dating. ML contributed to the evaluation of the results. ML, KN, and CS conceived of the presented idea. All authors provided comments to improve the manuscript.

**Data availability.**

At present, data can be obtained upon request. As agreed upon among the project participants, datasets will be made available to the public after the official completion of the Alpine Holocene Tree Ring Isotope Records (AHTRIR) project.

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





**Figure 1.** Correlations between TRW and stable isotopes calculated using the Spearman method in each age class (1-25, 26-50, 51-75, 76-
100, 101-200, 201-300, >300 yr) and with the total data of each species (blue box). Correlation results are displayed as columns. If the p-
value is > 0.05 (not significant), it is indicated in the graph as "ns". The upper panels a, b and c show correlations between the raw TRW
data; middle panels d, e and f show the correlations between the TRW and isotopes values. In the lower panels g, h and i show the correlations
in which the TRW values were detrended with a spline function. The values of larch trees are in red, those of cembran pine are in turquoise

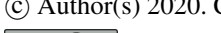



265

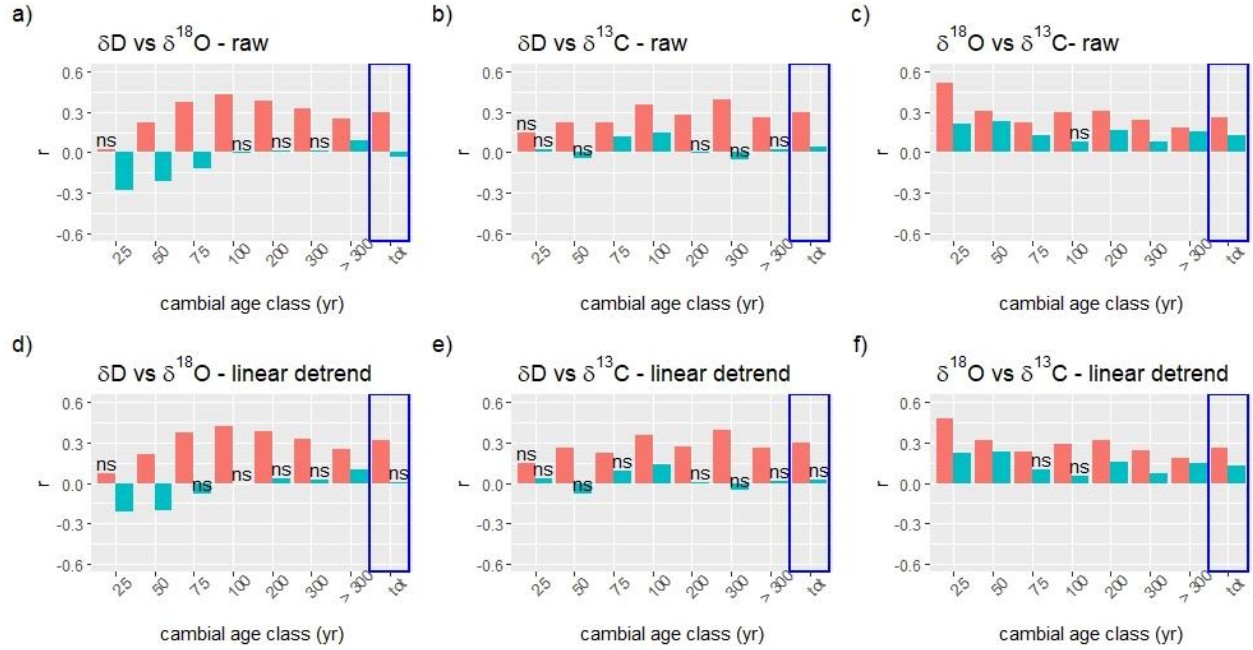

**Figure 2.** Correlations between the stable isotopes calculated using the Spearman method in each age class (1-25, 26-50, 51-75, 76-100, 101-200, 201-300, >300 yr) and also with the total data of each species (blue box). The p-values were calculated for each correlation displayed as a column, if P > 0.05, ns (non- significant) was indicated in the graph as NS. The upper panels a, b and c show the correlations between stable isotopes; middle panels d, e, f show the correlations between the isotope values after a linear detrending of each class. The values of larch trees are in red, those of cembran pine are in turquoise.



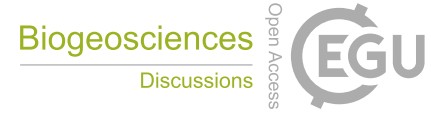

| Age class | N samples | | N trees | |
|---|---|---|---|---|
| | PICE | LADE | PICE | LADE |
| 1-25 | 124 | 78 | 35 | 25 |
| 26-50 | 225 | 171 | 54 | 40 |
| 51-75 | 323 | 230 | 70 | 50 |
| 76-100 | 334 | 264 | 73 | 57 |
| 101-200 | 1329 | 1048 | 85 | 68 |
| 201-300 | 896 | 776 | 68 | 57 |
| > 300 | 801 | 830 | 33 | 34 |
| tot | 4133 | 3471 | 108 | 84 |

**Table 1**. Number of isotope samples and number of trees of each cambial age class and in total.