# Peer review of "Cambial-age related correlations of stable isotopes and tree-ring widths in wood samples of tree-line conifers"

_Biogeosciences, 2020_

## Referee Comment (RC1) · Anonymous Referee #1 · 23 Dec 2020

General Comments. Thank you for the opportunity to review "Cambial-age related correlations of stable isotopes and tree-ring widths in wood samples of tree-line conifers" by Arosio, Ziehmer-Wenz, Nicolussi, Schlüchter, and Leuenberger, bg-2020-406. This study takes advantage of the large Alpine Holocene Tree-Ring Dataset to examine effects of age on the correlation between tree-ring width (TRW) and stable isotope ratios of hydrogen, oxygen and carbon. The manuscript extends an excellent study by a similar group of authors (Biogeosciences 17, 4871-4882, 2020) which found an influence of age on stable isotope ratios of oxygen and hydrogen but not carbon. Correlations between stable isotope ratios and TRW are relevant and within the scope of BG, but this paper seeks to explore the effect of age on these correlations, a narrow

and difficult subject given the known strong and non-linear effects of age on TRW. The dataset is apparently well-collected and impressive in its temporal coverage (9000 years) and replication (7604 samples). The approaches are standard, but the methods and assumptions appear to be valid and clearly outlined. The main conclusion, well supported by the data, is that the correlation between TRW and isotope ratios of hydrogen and oxygen is affected by age in the first 100 years. This is not surprising given that TRW is known to be strongly affected by age and that the same authors have already shown that isotope ratios of oxygen and hydrogen are affected by age. More detail in some of the methods would be helpful as described below. The authors give proper credit to related work, the title clearly reflects the contents of the paper, and the abstract is reasonably clear. The overall presentation is concise and well-structured, but problems with the English grammar make much of the writing unclear. Mathematical formulae, symbols, abbreviations and units are correctly defined and used. I have no suggestions for combining or eliminating major sections. The references and the supplementary material are appropriate.

Specific Comments. The authors find that the correlation between TRW and isotope ratios of hydrogen and oxygen, but not carbon, are affected by age in the first 100 years. These correlations are strongly affected by the method used to detrend the TRW data for age. More precisely there is a strong difference between the results of linear and spline detrending. The spline but not linear detrending is addressed in lines 143-144. The authors need to explain the difference between these two detrending approaches. Is it possible the cause of this difference is that in the linear case detrending was applied to both the isotope and TRW data, while in the spline case detrending was applied only to the TRW data?

For both species in the first 75 cambial years, the hydrogen isotope ratio is negatively correlated with TRW and the oxygen isotope ratio is positively correlated with TRW for both raw and linearly detrended data (Fig. 1, panels a, b, d, and e). The authors present an interesting and well-referenced explanation of this pattern in the Discussion.

On the other hand, this pattern leads to the expectation that there should be a negative correlation between the hydrogen and oxygen isotope ratios for both species over the same time period. Fig. 2 panels a and d show this for pine but not larch. Please explain.

Line 13 states "no trends" were found in adult trees. Please be more specific. Trends against what variable? Cambial age?

Line 55 states "To avoid a geographical effect that can lead to artificial trends, we have normalized the isotope values of each tree by subtracting the tree mean from each value". This is a reasonable approach as long as the variance does not change as a function of the mean. Is that the case?

Line 63 describes the linear detrending of the isotope and TRW values, but omits the variable against which this detrending was applied. Is this variable cambial age?

Line 67 says default settings were used for the spline detrending. Given that this detrending is central to the results, it would be useful to provide details of these settings. For example, what was the spline stiffness? Did you use the ratio or difference method to calculate residuals, etc.?

Line 95 states two patterns are different. How are they different?

Line 123 states "the correlation between the TRW and the two water isotopes of all the trees, not divided in cambial age classes, were non-significant." This is inconsistent with Figure 1 panels a, b, d, e, and f, which all show significant correlations between TRW and water isotopes where all age classes are combined.

Lines 124-126. Good point.

Technical Corrections.

Line 16. Replace "year" with "years".

Line 29 refers to "the two species" but the species are not introduced until the following

line.

Lines 32-33. Replace "along the cambial age" with "as a function of cambial age".

Line 34. Insert "have" before "analysed".

Line 46. Replace "spanning" with "combining" and insert "from individual trees" after "tree-rings".

Lines 75-79 simply explain the contents of Figure 1. This information belongs in the figure caption, not the Results Section.

Line 82. Replace "did neither change" with "changed neither"

Line 84. Replace "Similar is the pathway for correlations between TRW and d18O (Fig. 1 b,e) with positive" with "Similarly, TRW and d18O (Fig. 1 b,e) show positive".

Line 86. Replace "with respect to" with "than"

Line 128. Replace "are" with "is", delete "of", and replace "works" with "work".

Line 130. Replace "age" with "trees".

Line 134. Replace "was" with "has been".

Line 138. Insert "to" after "leading".

Line 143. Insert "the" before "growth"

Line 147. Replace "The correlation, between dD and d18O, changed during the juvenile phase in both phases with parallel trends..." with "That the correlation between dD and d18O changed during the juvenile phase in both species...".

Line 151. Replace "described before" with "already known".

---

## Referee Comment (RC2) · Anonymous Referee #2 · 7 Jan 2021

Review of "Cambial-age related correlations of stable isotopes and tree-ring widths in wood samples of tree-line conifers" by Tito Arosio et al.

This study presents the correlations between different tree ring parameters as tree-ring width (TRW) and stable isotopes ($\delta$D, $\delta$18O and $\delta$13C) from samples collected from the Eastern Alpine Conifer Chronology for different cambial-age groups.

The scientific output of this paper is very low. Since the authors already presented the cambial-age analysis in another paper, published last year in the same journal (doi.org/10.5194/bg-17-4871-2020), I did not understand why the authors did not include analyses presented here, in that paper and decided to publish them separately.

[Figure]

My feeling is just they want to artificially increase their number of publications, because the data and the scientific output presented here are not enough for an independent research paper, especially in such high impact journal as Biogeoscineces. More than that, the presented paper cannot be read independently, as an individual research paper, in order to understand which kind of data were used. To understand the actual age-related trends of the presented data, it is necessary to read another paper, of the same authors, which was published in the same journal.

The authors present the correlation between TRW and i) $\delta$18O, ii) $\delta$D and iii) $\delta$13C. Such kind of correlations are rather useless, first of all, due to the fact that the authors already presented in another paper the cambial age trend of these four different tree ring parameters and secondly, because between these parameters does not exist any links. The TRW does not influence $\delta$18O and $\delta$18O does not influence TRW, the same with other parameters. The variation of the TRW is independent of the $\delta$18O, $\delta$D, or $\delta$13C variations. The correlations are made between parameters that do not have a cause-and-effect relationship. When correlation analyses are performed, it is supposed to be a connection or a relationship between those two parameters, but in this case, the only connection can be the presence or absence of the trend in juvenile cambial age of the trees, and this aspect was showed in the previous paper.

The obtained correlations are due mainly to the trend of the data, and the trend is already a demonstrated fact of these series. When you correlate two data sets with similar or opposite trends, automatically you will get a correlation coefficient (positive or negative). And the explanation of such correlations, correlations between parameters that do not have a link between them, based on the physiological processes of trees are only speculations.

The paper even does not have a conclusion section. The last two sentences (three rows) of the discussion summarize the conclusions of the paper, but which does not bring anything new from the last published paper by the authors. (Line 158: In conclusion, our results confirm the existence of a juvenile phase in the $\delta$D and $\delta$18O

isotopes. . .).

The figures are of very poor quality. Moreover, the paper contains only 2 simple figures (bar figures), while the figures from supplementary contain too many very small figures, which are hard to follow.

Considering all the above-mentioned arguments, I conclude that the presented results of the paper present very low scientific significance and do not bring new/important scientific information, thus I recommend the paper to be rejected.

---

## Author Comment (AC1) · 25 Jan 2021

Reply to the BG the Anonymous Referee #1 :

In blue the reviewer's comments and suggestions, in black our answers, in red the sentence added to the revised manuscript text

*This study takes advantage of the large Alpine Holocene Tree-Ring Dataset to examine effects of age on the correlation between tree-ring width (TRW) and stable isotope ratios of hydrogen, oxygen and carbon. The manuscript extends an excellent study by a similar group of authors (Biogeosciences 17, 4871-4882, 2020) which found an influence of age on stable isotope ratios of oxygen and hydrogen but not carbon. Correlations between stable isotope ratios and TRW are relevant and within the scope of BG, but this paper seeks to explore the effect of age on these correlations, a narrow and difficult subject given the known strong and non-linear effects of age on TRW. The dataset is apparently well-collected and impressive in its temporal coverage (9000 years) and replication (7604 samples). The approaches are standard, but the methods and assumptions appear to be valid and clearly outlined. The main conclusion, well supported by the data, is that the correlation between TRW and isotope ratios of hydrogen and oxygen is affected by age in the first 100 years. This is not surprising given that TRW is known to be strongly affected by age and that the same authors have already shown that isotope ratios of oxygen and hydrogen are affected by age. More detail in some of the methods would be helpful as described below. The authors give proper credit to related work, the title clearly reflects the contents of the paper, and the abstract is reasonably clear. The overall presentation is concise and well-structured, but problems with the English grammar make much of the writing unclear. Mathematical formulae, symbols, abbreviations and units are correctly defined and used. I have no suggestions for combining or eliminating major sections. The references and the supplementary material are appropriate.*

We thank the reviewer for the kind and constructive comments.

*Specific Comments. The authors find that the correlation between TRW and isotope ratios of hydrogen and oxygen, but not carbon, are affected by age in the first 100 years. These correlations are strongly affected by the method used to detrend the TRW data for age. More precisely there is a strong difference between the results of linear and spline detrending. The spline but not linear detrending is addressed in lines 143-144.The authors need to explain the difference between these two detrending approaches. Is it possible the cause of this difference is that in the linear case detrending was applied to both the isotope and TRW data, while in the spline case detrending was applied only to the TRW data?*

We investigated the relationship between TRW and isotope in three different scenarios, (i) raw data,(ii) linear detrending of each age class of TRW and isotope series, and (iii) spline TRW detrending and raw isotopes. Scenario (iii) is the most used approach for climatic investigations.

In the first two scenarios, the variance of the values did not change after detrending, while in the spline detrend of TRW the variance changed, remaining constant through all the cambial age. The spline detrend of isotopes is not necessary because they do not change the variance in respect to the cambial age.

We included the following part in the revised text in the Material and method section, subsection 2.1:

"The p-values (statistical significance) were calculated for each correlation and are displayed as bars in Figure 1 and 2. If P < 0.05, they are labeled as "ns" (not significant) in the graphs. To identify possible correlations derived by common trends, we calculated the correlations in three different scenarios: (i) by using the raw values of isotopes and of TRW (Fig. 1 a,b,c and 2 a,b,c); (ii) by applying in R a linear  function to both the isotope and TRW values, separately for each cambial age class in function of the cambial age (Fig. 1d,e,f and 2d,e,f); and (iii) by using the raw,

un-detrended, isotope values and applying a spline detrending to all TRW values not subdivided into classes, as done normally in climatic studies (Fritts 1976) (Fig. 1g,h,i). Spline detrending is a common methodology used in dendroclimatology to remove the ageing signal. In this study we calculated it by applying a linear spline detrending function, using the default sets (using 67% of total series length and frequency response of 50%), to the raw TRW utilizing the R library dplR (Bunn 2008). A visual representation of the data in the three different scenarios is in the supplementary (fig. S3)."

We also added the following figure S3 in the supplementary:

[Figure]

**Figure S3.** TRW and d18O values during tree cambial age expressed in the three forms used for the correlation analysis. The upper panels show raw data of TRW (a) and $\delta^{18}O$ (b); the middle panels show linear detrended data of TRW (c) and $\delta^{18}O$ (d) ; the lower panels show linear spline detrended data of TRW (e) and raw data of $\delta^{18}O$ (f). The values of larch trees are in red, those of cembran pine are in turquoise.

*For both species in the first 75 cambial years, the hydrogen isotope ratio is negatively correlated with TRW and the oxygen isotope ratio is positively correlated with TRW for both raw and linearly detrended data (Fig. 1, panels a, b, d, and e). The authors present an interesting and well-referenced explanation of this pattern in the Discussion.*

*On the other hand, this pattern leads to the expectation that there should be a negative correlation between the hydrogen and oxygen isotope ratios for both species over the same time period. Fig. 2 panels a and d show this for pine but not larch. Please explain.*

We have shown before that deuterium fractionation in larch is different in respect to other conifers leading to a uniquely high depletion in Deuterium (Arosio et al, 2020, https://doi.org/10.3389/feart.2020.523073). Since the origin of this difference of larch is still unclear, we are presently unable to explain the correlation between $\delta D$ and $\delta^{18}O$ in larch. Yet, the interpretation presented in our manuscript fits well with cembran pine correlation.

We added the following sentence in the discussion:

"This interpretation also explains the negative correlation between the δD-δ$^{18}$O in the juvenile phase in cembran pine and the absence of correlation in the adult phase. In larch the correlations δD-δ$^{18}$O are different from those in cembran pine. Arosio et al. (2020b) showed that deuterium is uniquely depleted in larch, probably because of a different biological fractionation. Therefore, our interpretation does only hold Cembra pine. The correlation between the two isotopes in larch is presently difficult to interpret."

*Line 13 states "no trends" were found in adult trees. Please be more specific. Trends against what variable? Cambial age?*

We changed changed the sentence into:  "δD and δ$^{18}$O exhibited no age-related trends in adult trees older than 100 yr."

*Line 55 states "To avoid a geographical effect that can lead to artificial trends, we have normalized the isotope values of each tree by subtracting the tree mean from each value". This is a reasonable approach as long as the variance does not change as a function of the mean. Is that the case?*

In previous analysis we have shown that the geographical effect influences the mean but not the variance.  In any case, we tried to compute the analysis also with scaled values (correcting the data also with the variance), but this did not modify the results compared to the normalized results.

*Line 63 describes the linear detrending of the isotope and TRW values, but omits the variable against which this detrending was applied. Is this variable cambial age?*

Yes, we used the cambial age. We add this information to the sentence:
"by applying a linear detrend, in function of the cambial age, to the isotope and TRW values removing linear trends in each age class."

*Line 67 says default settings were used for the spline detrending. Given that this de-trending is central to the results, it would be useful to provide details of these settings. For example, what was the spline stiffness? Did you use the ratio or difference method to calculate residuals, etc.?*

We used the spline detrend function using 67% of total series length and frequency response of 50%. We tested different detrend parameters with minor differences, we also tried a negative exponential detrend that caused an increase of non-significant correlation. We added this information to the manuscript:

"Spline detrending is a common methodology used in dendroclimatology to remove the ageing signal. In this study we calculated it by applying a linear spline detrending function, using the default sets (using 67% of total series length and a frequency response of 50%), to the raw TRW utilizing the R library dplR (Bunn, 2008). "

*Line 95 states two patterns are different. How are they different?*

We change the sentence to:

"The correlations between the TRW and δ$^{13}$C have many age classes with no significant r including all values class. But the pattern for the spline-detrended (Fig. 1 i) values do not show negative r values in contrast to Fig. 1 c, f."

*Line 123 states "the correlation between the TRW and the two water isotopes of all the trees, not divided in cambial age classes, were non-significant." This is inconsistent with Figure 1 panels a, b, d, e, and f, which all show significant correlations between TRW and water isotopes where all age classes are combined.*

Thank you.  We change the sentence to:

"An additional result was that the correlation values between the TRW and the two water isotopes of all the trees, not divided in cambial age classes, were higher in cembran pine than in larch-"

*Lines 124-126. Good point.*

Thank you

*Technical Corrections.*
*Line 16. Replace "year" with "years".*

Corrected

*Line 29 refers to "the two species" but the species are not introduced until the following line.*

We added this information

*Lines 32-33. Replace "along the cambial age" with "as a function of cambial age".*

Done

*Line 34. Insert "have" before "analysed".*

Done

*Line 46. Replace "spanning" with "combining" and insert "from individual trees" after "tree-rings". Lines 75-79 simply explain the contents of Figure 1. This information belongs in the figure caption, not the Results Section.*

Done

*Line 82. Replace "did neither change" with "changed neither"*

Done

*Line 84. Replace "Similar is the pathway for correlations between TRW and d18O (Fig.1 b,e) with positive" with "Similarly, TRW and d18O (Fig. 1 b,e) show positive".*

Done

*Line 86. Replace "with respect to" with "than" Line 128. Replace "are" with "is", delete "of", and replace "works" with "work".*

Done

*Line 130. Replace "age" with "trees". Line 134. Replace "was" with "has been".*

Done

*Line 138. Insert "to" after "leading".*

Done

*Line 143. Insert "the" before "growth"*

Done

*Line 147. Replace "The correlation, between dD and d18O, changed during the juve-nile phase in both phases with parallel trends . . ." with "That the correlation between dDand d18O changed during the juvenile phase in both species . . .".*

Done

*Line 151. Replace "described before" with "already known".*

Done

---

## Author Comment (AC2) · 25 Jan 2021

Reply to the BG the Anonymous Referee #2:

In blue the reviewer's comments and suggestions, in black our answers, in red the sentence added to the revised manuscript text

Anonymous Referee #2

Review of "Cambial-age related correlations of stable isotopes and tree-ring widths in wood samples of tree-line conifers" by Tito Arosio et al.

*This study presents the correlations between different tree ring parameters as tree-ring width (TRW) and stable isotopes ($\delta D$, $\delta 18O$ and $\delta 13C$) from samples collected from the Eastern Alpine Conifer Chronology for different cambial-age groups. The scientific output of this paper is very low. Since the authors already presented the cambial-age analysis in another paper, published last year in the same journal (doi.org/10.5194/bg-17-4871-2020), I did not understand why the authors did not include analyses presented here, in that paper and decided to publish them separately.*

The analysis of the cambial age of the trees of the dataset have been analyzed before, showing that age trends are present at cambial age <100 yr, and not later. Here we used a different approach that is based on the analysis of statistical correlations between the different tree rings parameters aiming at obtaining data for the interpretation of the biological causes of the juvenile age trends. This approach was started after the publication of the previous paper and stimulated by more recent papers that study the influence of cambial metabolism on the cellulose isotopes. Therefore, this work is a new analysis how the TRW and cellulose isotopes are connected to each other in relation to age.

*My feeling is just they want to artificially increase their number of publications, because the data and the scientific output presented here are not enough for an independent research paper, especially in such high impact journal as Biogeoscineces.*

This is not the case, see statement to your previous remark.

*More than that, the presented paper cannot be read independently, as an individual research paper, in order to understand which kind of data were used. To understand the actual age-related trends of the presented data, it is necessary to read another paper, of the same authors, which was published in the same journal.*

It is stated in the abstract and introduction that this is a continuation of the previous paper published in Biogeoscience (Arosio et al, 2020a) and of that published in Frontier (Arosio et al, 2020b) by the same authors. Furthermore, we described the dataset used in the Materials and Methods section of this manuscript, therefore it is a stand-alone paper. In the present study, we analyze the correlations in relation to the cambial age and not the age-related trends.

*The authors present the correlation between TRW and i)$\delta 18O$, ii)$\delta D$ and iii)$\delta 13C$.Such kind of correlations are rather useless, first of all, due to the fact that the authors already presented in another paper the cambial age trend of these four different tree ring parameters*

That the "study of the correlations between TRW and the isotopes is useless" goes against recent literature, in which these correlations are more and more studied for a better understanding of the signals that the isotopes communicate. In the manuscript we quote several papers where the correlations are studied with contradictory results; moreover a very recent paper analyses the correlation difference between TRW and $\delta D$, linking them to different climatic conditions (Lehman et al. 2021 https://doi.org/10.1016/j.dendro.2020.125788). The correlations between TRW and carbon isotope were also studied (Shestakova et al, 2017, Shestakova et al, 2019). Even Biogeoscience recently published a paper where the authors used the correlations between TRW

and carbon isotopes (Deshpande et al. 2020 - https://doi.org/10.5194/bg-17-5639-2020). Our work increases the knowledge of this phenomenon, looking at how the correlations change according to cambial age and to various types of detrending methods. The relationship between the different isotopes in cellulose is also under investigation, and the combinations of them are used in paleoclimatic study (Nakatsuka et al. 2020, Xu et al 2020). Our data show that they can change as function of the cambial age. The correlations do not have transitive properties if R is < 1.0, so it is important to look not only at the correlations between TRW and isotopes but also among the isotopes.

We have added some sentences in the revised manuscript to underline the usefulness of the study of the relationships between TRW and isotopes:

"Only a few studies have analysed the correlations among TRW values and stable isotope fractionation with rather contrasting results, and some of them have used this relationship to extract climatic and physiological information (Deshpande et al. 2020, Lehmann et al 2021, Shestakova er al. 2017 Shestakova et al. 2019). Also, some combinations of the different isotopes have been used for paleoclimatic studies (Xu et al. 2020, Nakatsuka et al 2020). Most of these previous works used detrended TRW values and the calculated values of the correlation R factors were generally in the range 0.2 - 0.3 (Hafner et al. 2011; Kirdyanov et al. 2008; Schollaen et al. 2013; Shestakova et al. 2019; Weigl et al. 2007), but one work found correlations to be non-significant (Sidorova et al. 2010). However, none of the studies has taken into consideration how the cambial age of isotope samples can affect such correlation analyses. To fill this gap, we reanalyzed our database and divided the samples in classes of cambial age, with attention to the juvenile phase, i.e. the first 100 cambial years, and we studied the correlations between TRW and stable isotopes, using raw data as well as detrended data applying different detrending methods."

*and secondly, because between these parameters does not exist any links.*

In fact the presence of links among these parameters has been shown before. For example Szejner et al. (2020) indicated a "link between cambial activity and triose phosphate cycling", and TRW is known to be an index of cambial activity (Srivastava, 1976) and triose phosphate cycling is known to affect δ18O and δD values (Sternberg & DeNiro, 1983; Hill et al., 1995; Barbour & Farquhar, 2000, Augusti et al. 2006), in good agreement with the data we present.
Moreover, TRW and $\delta^{13}C$ are linked via carbon allocation strategies (Shestakova et al. 2019) and are affected by climatic and regional variations, but not by cambial age (our results).
In addition, we also studied the correlations among the isotopes sharing the same source, i.e. δD and $\delta^{18}O$. We found that the correlation between δD and $\delta^{18}O$ change with cambial age and they correlate differently for the two species (negatively for pine and positively for larch).

*The TRW does not influenceδ18O andδ18O does not influence TRW, the same with other parameters. The variation of the TRW is independent of the δ18O,δD, or δ13C variations.*

We agree that there is no direct causal influence of TRW on $\delta^{18}O$ and δD and viceversa, however these values may be influenced similarly by external or internal drives, and our work is focused to better understand these connections. A linkage between TRW and cellulose isotopes has been shown before, as stated above (Szejner et al. 2020, Shestakova et al 2019). This link can be caused by climatic driver or biochemical events, looking at our large database we are able to identify the biochemical linkage out to the common climatic driver. The aim work is to understand how these relationships change with respect to the cambial age, regardless of climatic factors.

*The correlations are made between parameters that do not have a cause-and-effect relationship. When correlation analyses are performed, it is supposed to be a connection or a relationship between those two parameters, but in this case, the only connection can be the presence or absence of the trend in juvenile cambial age of the trees, and this aspect was showed in the previous paper.*

The reviewer has to consider not only a time dependence (juvenile effect) but also a dependence on biochemical processes related to the metabolism. In order to disentangle these different influences we applied statistical methods. Correlation is a statistical measure of the strength of a relationship between two quantitative variables, and it does not imply causation! It is a tool aimed at identifying a common driver of two variables, and this may lead to novel interpretations. Our aim was to understand the change in relationship between the variables during cambial age for a better understanding of the causes of the age trend, that so far are just hypothesized. In the previous paper we did not analyse correlations and we did not propose interpretations for the juvenile age trends.

*The obtained correlations are due mainly to the trend of the data, and the trend is already a demonstrated fact of these series. When you correlate two data sets with similar or opposite trends, automatically you will get a correlation coefficient (positive or negative). And the explanation of such correlations, correlations between parameters that do not have a link between them, based on the physiological processes of trees are only speculations.*

We agree with the reviewer that correlations can be caused by common trends of the data. In fact, exactly to correct this problem, we made a linear detrend of each cambial age class ( fig 1 d,e,f and 2d,e,f). The finding that after detrending the correlations remained significant shows that they are not driven by the age-related trends of the juvenile phase. Moreover, we did not observe a correlation TRW- $\delta^{13}C$ in the juvenile phase using both raw and detrended data, even though we previously showed that carbon presents an age-related trend in cembran pine. Also, $\delta D$ has species-specific age trends but its correlation with TRW in the juvenile phase is not species-specific.

We add the figure S3 in supplement to better show the three different scenarios (raw, linear detrending, and spline for TRW):

[Figure]

**Figure S3.** TRW and d18O values during tree cambial age expressed in the three forms used for the correlation analysis. The upper panels show raw data of TRW (a) and d18O (b); the middle panels show linear detrended data of TRW (c) and d18O (d) ; the lower panels show linear spline detrended data of TRW (e) and raw data of d18O (f). The values of larch trees are in red, those of cembran pine are in turquoise

*The paper even does not have a conclusion section. The last two sentences (three rows) of the discussion summarize the conclusions of the paper, but which does not bring anything new from the last published paper by the authors. (Line 158: In conclusion, our results confirm the existence of a juvenile phase in the δD andδ18O isotopes...).*

We agree with reviewer and we significantly have rewritten the conclusion section:

"In conclusion, we studied the effect of cambial age on the modification of the relationships between tree rings components (TRW and cellulose isotopes). Our results indicate that during the juvenile phase the change of the δD and $\delta^{18}O$ isotopes of cellulose can be attributed to a high cambial activity that decreases with higher age. This indicates a different climatic sensitivity of the age groups, in accordance with Xu et al. (2020), implying that a simple detrending of the values of water isotopes may not be sufficient for paleoclimatic studies. The absence of a substantial change of trends between $\delta^{13}C$ and TRW indicates that plants do not modify the mechanisms of use of reserves as a function of age. We provide novel information for the use of the relationships between trees ring components (TRW and isotope compositions) to extract climatic information, in particular alerting readers that the cambial age and the TRW detrending methods can influence the results. The relationship between the isotopes of water (δD, $\delta^{18}O$) in cellulose shows a difference between juvenile phase and adult phase, but a different behavior is present in the two species. In adult phase there is a stronger correlation in larch than in cembran pine, in agreement with Arosio et al. (2020b)"

*The figures are of very poor quality. Moreover, the paper contains only 2 simple figures(bar figures), while the figures from supplementary contain too many very small figures, which are hard to follow.*

We have redrawn fig 1 and 2 to improve their quality. The supplementary have been simplified following reviewer's suggestion.